# Optimizing NPSB fertilizer rates for enhanced yield and yield components of mung bean (*Vigna radiata* L:) varieties

Daniel Manore[1]*, Hiwot Kelbo[1], Oumer Abdella[2], Francis Abuye[3]

**1** Wachemo University Department of Plant Science, Hossaena, Ethiopia, **2** Silte Zone Agricultural Office, Department of Field Crops, Worabe, Ethiopa, **3** Wachemo University Department of Natural Resource Management Hossaena, Ethiopia

* daniel.manore@wcu.edu.et

## Abstract

Ethiopia's mung bean sector faces profound constraints: persistently degraded soils, critically low adoption of essential NPSB fertilizers, and a severe shortage of improved varieties. These factors collectively cripple the crop's inherent productivity and national potential. To directly address these barriers, this two-year field study (2022 and 2023) evaluated the synergistic effects of improved mung bean varieties and NPSB fertilizer application on crop performance. The experiment was conducted using a factorial design within a randomized complete block layout, replicated 3 times, to test 3 key varieties: NVL-1, N-26, and Arkebe, at 5 different NPSB fertilizer rates of 0, 25, 50, 75, and 100 kg per hectare. Results decisively demonstrated that optimal growth and yield parameters were consistently achieved at the highest fertilizer levels (75 and 100 kg ha$^{-1}$). The N-26 variety emerged as the best across critical metrics, including plant height, branching, seed yield, and harvest index throughout both years. A standout performance occurred in 2022, where N-26 combined with 100 kg ha$^{-1}$ NPSB produced a peak grain yield of 1.94 t/ha. Arkebe's best yield was 1.78 t/ha at 75 kg ha$^{-1}$, higher than other varieties. Economic analysis further solidified N-26's superiority: paired with 100 kg ha$^{-1}$ NPSB, it delivered the highest net benefits, 47,704.17 ETB per hectare in 2022 and 49,856.85 ETB per hectare in 2023. Therefore, applying NPSB fertilizer at 100 kg ha$^{-1}$ to the N-26 variety is recommended to maximize mung bean productivity and profitability in the studied context.

## 1. Introduction

Beneath its unassuming appearance, the mung bean (*Vigna radiata*) emerges as a remarkable testament to resilience and nourishment [1,2]. As a proud member of the Papilionoideae legume family, it thrives in conditions where other crops falter. This resilient plant requires little water and flourishes in warm climates, performing a silent miracle beneath the surface: nitrogen fixation. By converting atmospheric nitrogen

**Data availability statement:** All relevant data are within the paper and its Supporting Information files.

**Funding:** The author(s) received no specific funding for this work.

**Competing interests:** The authors have declared that no competing interests exist.

into usable nutrients, it enriches the soil with approximately 40–47 kg of nitrogen per hectare, serving as a natural fertilizer for future crops [1,3].

It nourishes humans not only as protein-packed dry beans but also as crisp sprouts. Livestock benefit from it as forage, while the soil rejuvenates through its use as green manure [4]. Nutritionally, the mung bean is a heavyweight, boasting 24–26% protein, a wealth of essential amino acids, vitamins (A and C), and minerals such as iron, calcium, and zinc, all at a minimal cost [4]. Its composition—50.4% carbohydrates, 3.5–4.5% fiber, and just 1–3% fats—makes it a dietary cornerstone in the battle against malnutrition [5]. Beyond mere sustenance, it serves as a shield for health, offering benefits such as cholesterol reduction and blood sugar management [5].

In Ethiopia, the promise of the mung bean is palpable, yet perilously unfulfilled. Cultivated across 49,123 hectares, it yields 55,793 tons annually, a potential lifeline for food security. However, the harsh reality is stark: average yields stagnate at a meager 1.136 tons per hectare, a figure that pales in comparison to global standards [6].

The fertile promise of Dalocha District's fields is stifled by a silent crisis. Its soil cries out for nourishment [7]. Critical deficiencies in nitrogen (N), phosphorus (P), sulfur (S), and boron (B) cripple mung bean productivity, leaving plants starved, pods stunted, and farmers' hopes diminished [8]. The challenge is exacerbated by insufficient fertilizer use, limited knowledge among farmers of improved techniques, poor uptake of high-yielding crop varieties [9], and importantly, a substantial knowledge gap regarding the optimal application rates of NPSB fertilizer and the most effective crop varieties for local conditions [7,10]. Without this understanding, sustainable intensification remains out of reach. This study spans 2 growing seasons to examine how varying NPSB fertilizer rates influence mung bean growth, crop output, and ultimate yield in the Dalocha District, with the objective of determining the optimal application rate for maximum yield; determining the mung bean variety yielding the highest and most responsive results under these nutrient conditions; and assessing the economic feasibility and cost-effectiveness of using NPSB fertilizer for mung bean cultivation in the district, ensuring recommendations are both scientifically sound and financially prudent for farmers.

## 2. Materials and methods

### 2.1. Description of study area

Mataya Dange is located in the Dalocha district of the Silte zone. The region experiences a tropical wet and dry (savanna) climate. It is situated approximately 132 km southwest of Addis Ababa, nestled between the towns of Hosanna and Butejira.

### 2.2. Rainfall and temperature

The district's minimum and maximum temperatures are 14.21°C and 85°C, respectively, a mild climate suitable for agriculture. It receives about 161.3 mm (6.35 inches) of rainfall annually (Fig 1). Rainfall patterns are vital for residents' livelihoods, significantly impacting agricultural productivity and water availability.

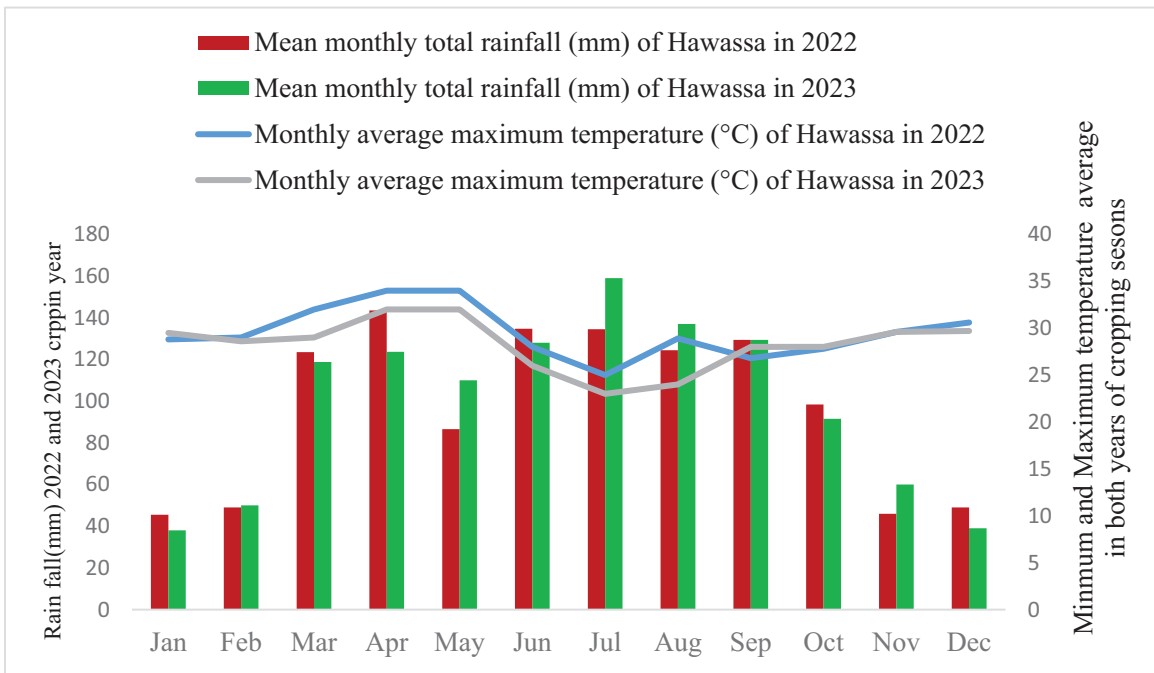

**Fig 1. Weather data of cropping years 'at Dalocha from Hawassa meteorological stations.**

## 2.3. Experimental materials description

The NPSB fertilizer utilized in the experiment consisted of the following nutrient composition: 18.9% nitrogen, 37.75% phosphorous, 6.95% sulfur, and 0.1% other elements. The selection of the NPSB fertilizer was based on soil fertility and fertilizer recommendations provided by Atlas [11]. The mung bean varieties employed in this experiment were obtained from the Melkassa Agricultural Research Center, shown in Table 1.

## 2.4. Treatments and experimental design

The experiment evaluated 3 Mung bean varieties (Table 1) combined with 5 NPSB fertilizer rates (0, 25, 50, 75, and 100 kg ha$^{-1}$), resulting in 15 treatments. Conducted over 2 cropping years (2021/22–2022/23), the study employed a factorial randomized complete block design (RCBD) with 3 replicates. Each 4 m² plot (2 m × 2 m) contained 6 rows, with 0.5 m and 1 m spacing between plots and blocks, respectively. The total experimental area was 296 m² (8 m × 37 m). Planting was done at 30 cm inter-row and 10 cm intra-row spacing, with seeds planted at a 3 cm depth. Two seeds were planted per hole, and NPSB fertilizer was applied using the drill method at sowing, placed 5 cm deep, and covered with soil. Uniform agronomic practices, including weeding, spacing, and disease management, were maintained across all plots.

**Table 1. Description of the experimental materials.**

| Varieties | Year of release | Seed size | Status | Source |
|---|---|---|---|---|
| N-26 | 2011 | Small | Released variety | Melkasa Agricultural Research Center. |
| Arkebe | 2014 | Small | Released variety | Humera Agricultural Research Center. |
| NVL-1 | 2014 | Small | Released variety | Melkasa Agricultural Research Center. |

## 2.5. Soil Physico-chemical properties

Before the sowing process began, soil samples were collected from 10 different locations at a depth of 0–30 cm using a zigzag sampling pattern. These individual samples were then mixed to form composite samples, ensuring a representative analysis of the soil. After the crops were harvested in both years of the study, soil samples were again collected from each treatment plot for further analysis. The soil properties analyzed included texture, pH, organic carbon (OC), total nitrogen (TN), available phosphorus (AP), cation exchange capacity (CEC), available sulfur (AS), and available boron (AB). These analyses were conducted at Wachamo University's soil and plant tissue analysis laboratory. Specifically, soil pH was measured potentiometrically in a 1:2.5 soil-water suspension [12]. Organic carbon (OC) was determined using the Walkley-Black method [13], while total nitrogen (TN) was analyzed via the Kjeldahl method [14]. Available phosphorus (AP) was measured using the Olsen method [15], and available sulfur (AS) was analyzed turbidimetrically [16]. Soil texture was determined using the Bouyoucos hydrometer method [17], and cation exchange capacity (CEC) was assessed using the ammonium acetate method [18]. Additionally, electrical conductivity (EC) was measured with an EC meter [18]. These comprehensive analyses were carried out both before planting and after harvest to monitor and evaluate changes in soil properties over the course of the study. This approach provided valuable insights into the impact of agricultural practices on soil health and fertility.

## 2.6. Data collection of growth and agronomic parameters

Days to emergence: The period from sowing until 50% of plants showed panicle tip emergence was recorded through visual observation.

Days to physiological maturity: The stage when 90% of plants in each plot reached physiological maturity was documented.

Nodule number per plant: At mid-flowering, 5 plants from the second inner rows were uprooted, washed gently, and separated into roots and shoots. Nodules were removed, spread on a sieve, and counted to determine the average number of nodules per plant.

Plant height (cm): The average height of 5 randomly selected plants (excluding roots) was measured at harvest using a meter scale.

Number of branches: Branches were counted at 90% maturity.

Leaf area index (LAI): Calculated as the total leaf area of 5 plants (cm²) divided by the land area they occupied.

Fresh and dry weight: Plant samples were patted, weighed for fresh weight, oven-dried, cooled in a desiccator, and reweighed for dry weight.

Number of Pods per Plant: Pods from 5 plants were counted to determine the average per plant.

Number of Seeds per Pod: Seeds in 10 randomly selected pods per treatment were counted for the mean.

Hundred-seed weight (g): A seed moisture content was adjusted to 11% by sun-drying, and then a 100-grain weight sample from each plot was weighed.

Aboveground Biomass (kg ha⁻¹): Dry biomass yield was recorded by harvesting 5 central rows per subplot, sun-drying, and converting to kg ha⁻¹.

Grain yield (kg ha⁻¹): Grains were weighed after threshing, cleaning, and sun-drying.

Harvest index (HI): Calculated as (Economic Yield/Biological Yield) × 100.

## 2.7. Data analysis

Prior to conducting the data analyses, we assessed the underlying assumptions regarding data distribution and carried out tests for multivariate normality, homogeneity of covariance matrices, and linearity. The Multivariate Analysis of Variance (MANOVA) analysis was carried out using SAS version 9.34 [19,20], following the procedures described by Gomez and Gomez [21]. Interpretations were then derived from the results. To compare the means of significant treatments, we applied a 5% probability level and used Tukey's HSD.

## 2.8. Partial budget analysis

A partial budget analysis [19,22] assessed the financial impact of treatments per hectare (ETB). It used average 2022–2023 grain and straw yields, valued at market prices. Input costs (planting) and output values (harvest) were included. Key variable costs covered NPSB fertilizer purchase, transport, and application; NPSB transport costs were derived from market/farm gate price differences. N, P, S B fertilizer costs used fixed enterprise prices. Actual grain and straw yields were reduced by 10% to reflect expected farm yields [22]. This adjusted data enabled calculation of gross field benefit, total variable costs, net benefit, and marginal rate of return.

## 3. Results

### 3.1. Soil analysis results

Table 2 presents soil physical and chemical properties of an experimental site before and after the harvest of a mung bean crop in 2 consecutive years, 2022 and 2023.

### 3.2. Key agronomic and phenological properties

The main effects and interaction effects are presented in detail in Table 3 below. The table provides a comprehensive breakdown of the statistical outcomes, highlighting the significance and magnitude of both the main effects of the independent variables and their potential interactions.

The effects of mung bean varieties and NPSB rates on yield and related traits are presented in Table 4. This table highlights the interactions among cropping year, variety, and NPSB fertilizer rates on mung bean yield and its components.

## 4. Discussions

### 4.1. Soil physico-chemical properties

The study showed changes in the soil properties after harvest (Table 2). The sand content decreased slightly, while the silt content increased in both years, indicating reduced soil coarseness due to root activity and organic matter decomposition [23]. The clay content remained stable, maintaining the soil structure. Soil pH increased following NPSB fertilizer application, rising from 5.63 to 5.8 in 2022 and to 6.0 in 2023, highlighting its role in neutralizing soil acidity [23]. The organic carbon (OC) content rose from 2.8% to 3.5% in 2022 and further to 3.64% in 2023, reflecting the contribution of mung bean to the organic matter [24]. Total nitrogen increased from 0.06 ppm to 0.14 ppm in 2022 and to 0.17 ppm in 2023, attributed to atmospheric nitrogen fixation by mung beans [25,26].

Available phosphorus (Avail P) increased from 27.25 ppm to 29.92 ppm in 2022 and to 30.7 ppm in 2023, linked to enhanced phosphorus availability from fertilizer [26]. The sulfur (S) content increased from 18.25 ppm to 20.49 ppm in 2022 and to 20.6 ppm in 2023, consistent with the use of NPSB fertilizer [27,28]. The boron (B) content increased from 0.57 ppm to 0.66 ppm in 2022 and to 0.8 ppm in 2023 due to fertilizer application [29]. The cation exchange capacity (CEC)

Table 2. Soil physical and chemical properties of the experimental site.

| Cropping year | Soil tests | Physical properties (%) | | | PH(1:2.5 H2O) | OC (%) | Total N (%) | P(ppm) | S (ppm) | B(ppm) | CEC [cmol(+)kg$^{-1}$] |
|---|---|---|---|---|---|---|---|---|---|---|---|
| | | Sand | Silt | Clay | | | | | | | |
| 2022 | Before planting | 50 | 18 | 32 | 5.63 | 2.8 | 0.06 | 27.25 | 18.25 | 0.57 | 22.22 |
| | After harvesting | 49.62 | 18.1 | 32.29 | 5.8 | 3.5 | 0.14 | 29.92 | 20.49 | 0.66 | 22.78 |
| 2023 | Before planting | 49.62 | 18.1 | 32.29 | 5.8 | 3.5 | 0.14 | 29.92 | 20.49 | 0.66 | 22.88 |
| | After harvesting | 50 | 20.3 | 29.7 | 6 | 3.64 | 0.17 | 30.7 | 20.6 | 0.8 | 29 |

Table 3. The main and interaction effects of different parameters are presented in detail in Table 3 below.

| Treatment | Nodule number per plant | Days to flowering | plant height (cm) | Number of branches | pods per plant | seed per pod | Fresh weight (gm) per plant | Dry matter (gm) per plant | Leaf area index | Days to 90% maturity | 100 seed weight (g) | Biological yield (t ha$^{-1}$) | Grain yield (t ha$^{-1}$) | Harvest index |
|---|---|---|---|---|---|---|---|---|---|---|---|---|---|---|
| **Cropping year(CR)** | | | | | | | | | | | | | | |
| 2022 | 8.8 | 59.5[b] | 52.23[b] | 5.43[b] | 27.4[b] | 10.5[b] | 27.2 | 2.3b | 0.27[b] | 78.5[b] | 50.63[b] | 4.23 | 1.58[b] | 0.443[b] |
| 2023 | 8.4 | 63.33[a] | 58.2[a] | 5.6[a] | 32.0[a] | 12.4[a] | 28.6 | 2.6a | 0.32[a] | 79.6[a] | 55.2[a] | 4.25 | 1.7[a] | 0.521[a] |
| LSD (5%) | 1.8 | 3.91 | 6.3 | 0.9 | 4.5 | 2.2 | ns | 1.1 | 0.26 | 3.6 | 3.6 | ns | 0.45 | 0.2 |
| CV(%) | 8.5 | 4.09 | 6.5 | 9.4 | 12 | 4.5 | 4.9 | 1.3 | 0.16 | 8.6 | 7.8 | 2.1 | 0.8 | 0.35 |
| **Varieties (V)** | | | | | | | | | | | | | | |
| NVL-1 | 8.38[b] | 45.67[b] | 52.2[bc] | 5.4[c] | 27.3[b] | 13.3[b] | 29.6[b] | 4.6[b] | 0.33[b] | 78.4[b] | 55.93[b] | 4.27[b] | 1.76[b] | 0.521[B] |
| N-26 | 8.45[b] | 44.5[b] | 53.27[b] | 5.6[b] | 29.0[a] | 14.8[a] | 32.6[a] | 5.2[a] | 0.34[b] | 79.76[a] | 57.0[a] | 4.42[a] | 1.95[a] | 0.536[a] |
| Arkebe | 9.6[a] | 50.6[a] | 58.53[a] | 5.97[a] | 30.0[a] | 12.5[b] | 28.6[b] | 4.8[b] | 0.38[a] | 78.9[a] | 55.2[b] | 4.099[c] | 1.74[b] | 0.526[a] |
| LSD (5%) | 1.94 | 4.7 | 6.8 | 1.5 | 3.2 | 1.7 | 3.5 | 1.5 | 0.14 | 6.2 | 3.8 | 2.2 | 0.48 | 0.12 |
| CV(%) | 9.6 | 6.4 | 9.3 | 12.5 | 8 | 5.3 | 9.8 | 2.6 | 0.23 | 11 | 8.4 | 1.3 | 0.83 | 0.4 |
| **NPSB(kg ha$^{-1}$)** | | | | | | | | | | | | | | |
| 0 | 8.38[d] | 45.67[c] | 52.2[c] | 5.1[c] | 24.5[c] | 12.63[b] | 13.3[e] | 2.5 cd | 0.29[bc] | 78.4[d] | 47.78[d] | 3.37 cd | 1.20[d] | 0.231[d] |
| 25 | 9.6[c] | 60.6[b] | 58.53[b] | 5.27[c] | 26.3[b] | 12.5[b] | 18.3[b] | 2.7c | 0.29[bc] | 78.9[c] | 52.4[c] | 3.79[c] | 1.28[d] | 0.301[c] |
| 50 | 14.7[b] | 61.5[a] | 62.5[a] | 5.6[b] | 29.0[a] | 14.5[a] | 19.6[b] | 2.8c | 0.32[b] | 80.76[b] | 55.4[bc] | 4.05[b] | 1.48[c] | 0.341[c] |
| 75 | 21.4[a] | 60[b] | 60.4[a] | 6.77[a] | 31.4[a] | 14.5[a] | 26[a] | 4.8[ab] | 0.35[b] | 80.3[b] | 54.73[ba] | 4.3[a] | 1.62[b] | 0.47[b] |
| 100 | 20.77[a] | 63[a] | 62.5[a] | 6.0[a] | 30.0[a] | 14.2[a] | 32.6[a] | 5.4[a] | 0.48a | 84.4[a] | 57.6[a] | 4.4[a] | 1.94[a] | 0.54[a] |
| LSD (5%) | 2.5 | 4.9 | 7.2 | 1.7 | 3.6 | 2.1 | 7.6 | 1.9 | 0.18 | 6.9 | 4.2 | 1.4 | 0.51 | 0.23 |
| CV(%) | 9.4 | 6.8 | 9.9 | 15.5 | 7.7 | 4.4 | 13.7 | 3.1 | 0.36 | 12.6 | 8.9 | 1.9 | 0.78 | 0.57 |
| **P statistics** | | | | | | | | | | | | | | |
| CR | ns | 0.0033 | 0.022 | <.0001 | 0.023 | 0.034 | ns | 0.023 | 0.04 | 0.0601 | 0.024 | ns | 0.002 | 0.034 |
| Varieties | 0.004 | <.0001 | 0.1456 | <.0003 | 0.045 | 0.025 | 0.008 | 0.015 | 0.026 | <.0001 | 0.013 | 0.04 | 0.024 | 0.001 |
| NPSB (kg ha$^{-1}$) | 0.023 | <.003 | 0.0431 | 0.0134 | 0.0034 | 0.032 | 0.028 | 0.006 | 0.011 | 0.026 | 0.006 | 0.011 | 0.025 | <.0001 |
| CR* V | ns | ns | ns | ns | ns | ns | ns | ns | ns | ns | ns | ns | ns | ns |
| CR* Fertilizer | ns | ns | ns | ns | ns | ns | ns | ns | ns | ns | ns | ns | ns | ns |
| V* Fertilizer | ns | ns | ns | ns | 0.0424 | 0.034 | ns | ns | 0.004 | ns | 0.021 | 0.005 | 0.032 | ns |
| CR* V* NPSB | ns | ns | ns | ns | ns | ns | 0.01 | 0.021 | 0.0023 | ns | ns | ns | ns | 0.001 |

*Letters in the same vertical column indicate no significant differences (p ≤ 0.05, 0.01, or 0.001). CV is the coefficient of variation, LSD denotes least significant differences, and ns means non-significant.*

**Table 4. Two-way interaction effects are shown for different yield components on mung bean.**

| Variety Type | NPSB rates (kg ha⁻¹) | Number of pods per plant | Number of seeds per pod | Number of seeds per plant | Hundred seed weight (g) | Biological yield(t ha⁻¹) | Grain yield(t ha⁻¹) |
|---|---|---|---|---|---|---|---|
| NVL-1 | 0 | 26.5[hf] | 10.5 [ef] | 278.56[gh] | 49.9[igh] | 3.5742[efg] | 1.08e[h] |
| NVL-1 | 25 | 28.4[ef] | 11.3[fh] | 299.5[ih] | 51.2[ih] | 3.397.1[fg] | 1.84[ih] |
| NVL-1 | 50 | 33.0[ef] | 10.53[fe] | 338.66[egh] | 52.59[fgh] | 3.860[efg] | 1.28[feh] |
| NVL-1 | 75 | 39.77[bd] | 12.33[ed] | 464.5 fg | 50.67[edcf] | 4.1691[abc] | 1.52[bc] |
| NVL-1 | 100 | 48.54[ba] | 12.5[cb] | 660.6[ab] | 55.93[bac] | 4.2603[ab] | 1.76[b] |
| N-26 | 0 | 29.34[h] | 11.5 fg | 288.65[ifg] | 47.78[fh] | 3.3685[hgi] | 1.20[h] |
| N-26 | 25 | 33.45[e] | 12.987 cd | 339.6[gh] | 52.4[ecd] | 3.7824[efg] | 1.28[fgh] |
| N-26 | 50 | 38.56[ce] | 12.77 [ac] | 472.4[dfeh] | 55.4[ba] | 4.0432[bc] | 1.48 cd |
| N-26 | 75 | 45.17[ba] | 14[b] | 560.2[c] | 54.73[bc] | 4.2597[ba] | 1.6[b] |
| N-26 | 100 | 51.8[a] | 15.8[a] | 769.7a | 57.0[a] | 4.4114[a] | 1.94[a] |
| Arkebe | 0 | 26.77[fh] | 8.9[h] | 267.1[ij] | 46.3[ij] | 3144.3[ik] | 1.27[gh] |
| Arkebe | 25 | 32.64[ef] | 11.12[fe] | 322.45[ihj] | 47.77[fgh] | 3.5588[gh] | 1.285 [fdh] |
| Arkebe | 50 | 34.65[de] | 12.6 cd | 414.0[gfh] | 51.6[dfe] | 3.7887[fgh] | 1.38[de] |
| Arkebe | 75 | 42.34[bd] | 12.48[cb] | 536.67[bdc] | 50.63[bed] | 4.1248[cde] | 1.78[b] |
| Arkebe | 100 | 46.47[ba] | 14.5a[cb] | 492.44[dce] | 55.2[bac] | 4.0985[bde] | 1.7[acb] |
| **LSD (5%)** | | **6.2** | **2.3** | **65** | **3.6** | **2.085** | **0.87** |
| **CV(%)** | | **10.7** | **11.5** | **15.5** | **8.5** | **12.2** | **3.8** |

*Letters in the same vertical column indicate no significant differences (p ≤ 0.05, 0.01, or 0.001). CV is the coefficient of variation, and LSD denotes significant least differences.*

improved significantly, rising from 22.22 to 22.78 cmol(+)/kg in 2022 and to 29 cmol(+)/kg in 2023, enhancing the soil nutrient retention.

## 4.2. Effects on growth and phenology

**4.2.1. Effects on nodule number per plant.** No significant differences in nodule numbers were observed between 2022 and 2023. However, Arkebe exhibited the highest nodule count (9.6), followed by NVL-1 (8.38) and N-26 (8.45), demonstrating genetic variability in nodulation capacity [30]. Nodule numbers peaked at 75 kg ha⁻¹ (21.4) and slightly decreased at 100 kg ha⁻¹ (20.77), suggesting that improved phosphorus availability enhances nodulation [30,31], as shown in Table 3.

**4.2.2. Effects on days to flowering.** Significant variations in flowering duration were observed between the 2 years, with delayed flowering in 2023 (63.33 days) compared to 2022 (59.5 days), likely due to environmental factors such as temperature and rainfall. [32] Table 3. Among the genotypes, Arkebe exhibited the longest flowering duration (50.6 days), followed by NVL-1 (45.67 days) and N-26 (44.5 days), reflecting genetic differences in phenological responses. Higher NPSB application rates delayed flowering, with 50 kg ha⁻¹ (61.5 days) and 100 kg ha⁻¹ (63 days) showing the longest durations, consistent with findings by [33].

**4.2.3. Effects on plant height.** *Plant* height was significantly greater in 2023 (58.2 cm) than in 2022 (52.23 cm), likely due to improved growing conditions or enhanced soil fertility (Table 3). Among the varieties, Arkebe was the tallest (58.53 cm), followed by N-26 (53.27 cm) and NVL-1 (52.2 cm), demonstrating genetic variability in plant architecture [6,34]. Phosphorus in NPSB fertilizer improved root growth, nutrient uptake, and drought tolerance, contributing to increased plant height [35,36].

**4.2.4. Effects on the number of branches per plant.** Branching increased significantly in 2023 (5.6) compared to 2022 (5.43), with Arkebe exhibiting the highest branching capacity (5.97). Branching peaked at 6.77 with 75 kg ha$^{-1}$ NPSB application, indicating that balanced nutrient application enhances branching. This aligns with [37], who found that increased nutrient availability promotes branch formation and vigorous growth.

**4.2.5. Effects on days to 90% maturity.** The 2023 season required slightly more time to reach maturity compared to 2022. Varietal differences were significant, with N-26 and Arkebe taking longer than NVL-1. Higher NPSB application rates delayed maturity, with 100 kg ha$^{-1}$ resulting in the longest duration (84.4 days) (Table 3). This delay is likely due to nitrogen-promoting vegetative growth, which postpones reproductive development, consistent with the findings of [38].

## 4.3. Effects on the yield components and yield

**4.3.1. *Effects on the number of pods per plant.*** The combined data from 2022 and 2023 revealed a significant interaction between variety and fertilizer level (p ≤ 0.05), as shown in Tables 3 and 4. The N-26 variety outperformed NVL-1 and Arkebe at 100 kg ha$^{-1}$, although the differences were minimal at lower fertilizer rates. This superior performance is likely due to better nutrient uptake, which is consistent with the findings of [39]. The study aligns with [40] emphasis on trait selection in breeding and [41]'s focus on matching varieties with appropriate fertilizer management. N-26 produced the highest number of pods (51.8) at 100 kg ha$^{-1}$, supporting [42] research on the role of nitrogen in legume pod development.

**4.3.2. *Effects on the number of seeds per pod.*** The number of seeds per pod increased with fertilizer application, but the extent varied by variety. N-26 recorded the highest number (15.8 at 100 kg ha$^{-1}$; Table 4), indicating superior reproductive efficiency under high nutrient conditions. These results highlight the importance of variety-fertilizer combinations for yield optimization, which is consistent with crop science principles [43]. The findings support [44] and [45] in emphasizing multifactorial management for sustainability.

**4.3.3. *Effects on seeds per plant.*** This metric combines pods per plant and seeds per pod, providing a holistic view of yield. N-26 consistently outperformed other varieties, especially at higher fertilizer rates (Table 4), demonstrating its potential for high-yield systems. The strong response of N-26 to fertilizer aligns with [46] and [45] While Arkebe showed stable performance across fertilizer rates, its yield potential remained lower than that of N-26 and NVL-1.

**4.3.4. Effects on hundred seed weight (g).** N-26 exhibited higher seed weights across all fertilizer rates, particularly at 100 kg ha$^{-1}$ (57.0 g; Table 4). Arkebe recorded the lowest weight at 0 kg ha$^{-1}$ (46.3 g) but improved significantly at 100 kg ha$^{-1}$ (55.2 g), indicating greater responsiveness to fertilizer than NVL-1, consistent with [47]. Increasing fertilizer rates generally enhanced seed weight across all varieties, aligning with [35], who reported that NPSB fertilization boosts seed weight.

**4.3.5. Effects on biological yield (t/ha).** N-26 attained the highest biological yield at 100 kg ha$^{-1}$ (4.4114 t/ha). NVL-1 showed a moderate increase (3.5742 t/ha to 4.2603 t/ha), while Arkebe exhibited a more significant response (3.1443 t/ha to 4.0985 t/ha). These results emphasize the importance of optimal nutrient management, as 100 kg ha$^{-1}$ was highly effective for N-26 [48,49]. N-26 demonstrates strong potential for high productivity, while Arkebe is less resilient under low-input conditions, likely due to environmental factors [26,50].

**4.3.6. Effects on grain yield (t/ha).** N-26 achieved the highest grain yield at 100 kg ha$^{-1}$ (1.94 t/ha), followed by Arkebe (1.7 t/ha) and NVL-1 (1.76 t/ha; Table 4). These results align with [51], indicating treatment effects rather than random variation. Grain yield increased with fertilizer application, but the response was nonlinear. For instance, NVL-1 showed a yield decline at 50 kg ha$^{-1}$ (1.28 t/ha) compared with that at 25 kg ha$^{-1}$ (1.84 t/ha), suggesting a potential threshold effect or nutrient imbalance at intermediate rates. Optimal fertilizer application likely provided the best nutrient balance for growth, maximizing yield [52]. These findings underscore the importance of proper fertilizer use for enhancing crop yield [53]. Varieties responded differently to fertilizer rates, with N-26 demonstrating greater efficiency in converting nutrients into grain yield.

**4.3.7. Effects on fresh and dry weight.** The three-way interaction effects of years, varieties, and NPSB rates were highly significant (p ≤ 0.01), as shown in Tables 3 and 5. Variety N-26 consistently outperformed NVL-1 and Arkebe,

**Table 5. Interaction effects are shown for different yield components.**

| Treatment combinations | | Fresh weight (gm) per plants | | Dry matter (gm) per plants | | LAI | | Harvest index | |
|---|---|---|---|---|---|---|---|---|---|
| Varieties | NPSB rates (kg ha$^{-1}$) | 2022 | 2023 | 2022 | 2023 | 2022 | 2023 | 2022 | 2023 |
| NVL-1 | 0 | 11.3$^e$ | 15.4$^d$ | 1.9$^d$ | 2.1$^d$ | 0.22$^d$ | 0.25$^c$ | 0.211$^d$ | 0.232$^d$ |
| | 25 | 15.6$^d$ | 21.5$^{ab}$ | 2.3$^c$ | 2.4$^c$ | 0.23$^d$ | 0.25$^c$ | 0.311$^{bc}$ | 0.355$^c$ |
| | 50 | 18.6$^c$ | 19.7$^{ab}$ | 2.6$^b$ | 2.4$^c$ | 0.28$_b$ | 0.283$^{bc}$ | 0.388$^{ab}$ | 0.398$^{ab}$ |
| | 75 | 23.6$^{ab}$ | 28.5$^a$ | 3.2$^b$ | 4.3$^b$ | 0.27$^b$ | 0.35$^b$ | 0.343$^{ba}$ | 0.384$^{ab}$ |
| | 100 | 29.6$^a$ | 30.3$^a$ | 4.6$^a$ | 4.8$^{ab}$ | 0.32$^a$ | 0.45$^a$ | 0.521$^a$ | 0.571$^a$ |
| N-26 | 0 | 13.3$^e$ | 16.8$^d$ | 2.2 cd | 2.5$^c$ | 0.24$^c$ | 0.29$^{bc}$ | 0.211$^d$ | 0.256$^c$ |
| | 25 | 18.3$^b$ | 23.8$^{ab}$ | 2.4$^c$ | 2.7$^{cb}$ | 0.25$^c$ | 0.29$^{bc}$ | 0.301$^b$ | 0.365$^{bc}$ |
| | 50 | 19.6$^b$ | 20.7$^{ab}$ | 2.7$^b$ | 2.8$^{cb}$ | 0.29$^b$ | 0.32$^b$ | 0.341$^b$ | 0.386$^{ba}$ |
| | 75 | 26$^a$ | 27.5$^a$ | 3.7$^{ab}$ | 4.8$^{ab}$ | 0.27$^b$ | 0.35$^b$ | 0.465$^{ab}$ | 0.495$^{ab}$ |
| | 100 | 32.6$^a$ | 34.6$^a$ | 4.5$^a$ | 5.5$^a$ | 0.35$^a$ | 0.48a | 0.54$^a$ | 0.589$^a$ |
| Arkebe | 0 | 11.0$^e$ | 15.3$^d$ | 1.7$^d$ | 1.9$^d$ | 0.22$^d$ | 0.24$^d$ | 0.248$^d$ | 0.248$^c$ |
| | 25 | 16.6$^d$ | 19.5$^{ab}$ | 2.4c | 2.5$^c$ | 0.21$^d$ | 0.23$^d$ | 0.321$^b$ | 0.353$^c$ |
| | 50 | 18.6$^b$ | 19.6$^{ab}$ | 2.6$^b$ | 2.6$^c$ | 0.27$^b$ | 0.30$_b$ | 0.341$^b$ | 0.364$^b$ |
| | 75 | 27.2$^a$ | 28.5$^a$ | 3.2$^b$ | 4.2$^b$ | 0.29$^b$ | 0.37$^b$ | 0.465$^{ab}$ | 0.483$^{ab}$ |
| | 100 | 28.6$^a$ | 30.6$^a$ | 4.8$^a$ | 5.2$^a$ | 0.38$^a$ | 0.46$^d$ | 0.526$^a$ | 0.554$^a$ |
| LSD (5%) | | **10.3** | 9.8 | **2.1** | 1.97 | **0.197** | **0.22** | **0.26** | **0.23** |
| CV(%) | | **23.6** | 19.4 | **4.3** | 3.9 | **0.42** | **0.38** | **0.62** | **0.57** |

*Letters in the same vertical column indicate no significant differences (p ≤ 0.05, 0.01, or 0.001). CV is the coefficient of variation, LSD denotes significant differences, and ns means non-significant.*

particularly at higher NPSB rates (e.g., 34.6 gm fresh weight at 100 kg ha$^{-1}$ in 2023, compared to 30.3 gm and 30.6 gm for NVL-1 and Arkebe, respectively). Arkebe, while underperforming at lower NPSB rates, showed greater responsiveness at higher rates (75 and 100 kg ha$^{-1}$), nearly matching N-26 at 100 kg ha$^{-1}$. Across all varieties, fresh weight increased with higher NPSB rates, with NVL-1 showing a rise from 11.3 gm at 0 kg ha$^{-1}$ to 29.6 gm at 100 kg ha$^{-1}$ in 2022 and from 15.4 gm to 30.3 gm in 2023. The optimal NPSB rate for maximizing the fresh weight was consistently 100 kg ha$^{-1}$ across all varieties and years, as supported by [54,55]. Similarly, dry matter yields improved with increasing NPSB rates, such as NVL-1's increase from 1.9 gm/plant (0 kg ha$^{-1}$) to 4.6 gm/plant (100 kg ha$^{-1}$) in 2022, with similar trends for N-26 and Arkebe. Genetic differences in nutrient use efficiency were evident, as Arkebe, despite lower yields at 0 kg ha$^{-1}$, matched N-26 at 100 kg ha$^{-1}$ [56]. Overall, yields were higher in 2023 than in 2022, likely due to favorable environmental conditions like rainfall and temperature, which enhanced nutrient availability [57]. Balanced fertilization (NPSB: nitrogen, phosphorus, sulfur, and boron) significantly promoted growth and biomass [58].

**4.3.8. Effects on the leaf area index.** The LAI values vary significantly across varieties, NPSB levels, and years, indicating a complex interaction (Table 5). For instance, NVL-1's LAI at 100 kg ha$^{-1}$ NPSB increased from 0.32 in 2022 to 0.45 in 2023, suggesting year-specific environmental or management influences. Generally, LAI increases with higher NPSB levels, peaking at 100 kg ha$^{-1}$, reflecting improved nutrient availability for photosynthesis and growth. However, the response magnitude varies by variety and year. N-26 showed a smaller LAI increase (0.27 to 0.35) from 75 kg ha$^{-1}$ to 100 kg ha$^{-1}$ in 2022 compared to Arkebe (0.29 to 0.38). Arkebe consistently achieved the highest LAI at 100 kg ha$^{-1}$ NPSB, indicating superior responsiveness, while NVL-1 and N-26 exhibited lower values, likely due to genetic differences in nutrient use. Increased LAI is linked to enhanced nutrient availability, supporting physiological processes [59], consistent with studies emphasizing nutrient importance in plant development [5,60].

**4.3.9. Effects on the harvest index.** NVL-1 and N-26 demonstrate a significant increase in harvest index (HI) (Table 5) with rising NPSB rates, peaking at 100 kg ha$^{-1}$ in both years (NVL-1: 0.521 in 2022, 0.571 in 2023; N-26: 0.54 in 2022, 0.589 in 2023). Arkebe also responded positively but shows a less pronounced HI increase, especially at lower NPSB rates (25 and 50 kg ha$^{-1}$). Across all varieties and years, HI consistently rises with higher NPSB rates, reaching its maximum at 100 kg ha$^{-1}$, indicating improved biomass partitioning into economic yield, aligning with [33,61]. The lowest HI values occur at 0 kg ha$^{-1}$ NPSB, highlighting the critical role of fertilization in resource allocation for grain production. Higher HI values in 2023 compared to 2022 suggest more favorable environmental conditions, potentially due to variations in rainfall, temperature, or other agronomic factors [57].

This study found a strong positive correlation between fertilizer application and nodule number per plant ($R^2 = 0.73$, $p < 0.01$), highlighting that increased fertilizer application enhances nodule formation, essential for nitrogen fixation, consistent with [62]. Similarly, fertilizer application strongly correlated with grain yield ($R^2 = 0.95$, $p < 0.001$), indicating its role in boosting yield and economic returns, aligning with [63]. A moderate positive correlation was observed between seeds per pod and seed weight ($R^2 = 0.85$, $p < 0.01$), suggesting that more nodules improve seed weight, a critical yield factor [62]. Additionally, plant height and branch number showed a high positive correlation ($R^2 = 0.84$, $p < 0.05$), implying that taller plants with more branches may enhance yield and reduce weather-related risks.

## 4.4. Partial budget analysis

The results revealed that the N-26 variety, fertilized with NPSB at 100 kg ha$^{-1}$, generated the highest net benefit, yielding 47,704.17 ETB/ha in 2022 and 49,856.85 ETB/ha in 2023. These findings are consistent with those of [64,65], who also demonstrated that optimal NPSB fertilizer application enhances the economic performance of mung bean production.

## 5. Conclusion

The NPSB fertilizer rates of 75 and 100 kg ha$^{-1}$ were found to result in the highest growth and yield parameters. Among the varieties, N-26 excelled in branching, height, seed yield, and harvest index over the 2 years. In 2023, N-26 with 100 kg ha$^{-1}$ produced the highest grain yield of 1.94 t/ha, while Arkebe yielded the lowest at 1.78 t/ha, with 75 kg ha$^{-1}$. The N-26 variety also achieved the greatest net benefit, yielding 47,704.17 ETB per hectare in 2022 and 49,856.85 ETB/ha in 2023 with a 100 kg ha$^{-1}$ application. It is advisable to use 100 kg ha$^{-1}$ of NPSB fertilizer for the N-26 variety and 75 kg ha$^{-1}$ for Arkebe. Further research is needed to confirm these results over multiple years and across additional varieties and fertilizer rates.

## Acknowledgments

We express our gratitude to Wachemo University and the Dalocha Woreda Silete zone for providing us with the necessary field and laboratory facilities.

## Author contributions

**Data curation:** Hiwot Kelbo, Oumer Abdella.

**Methodology:** Hiwot Kelbo, Oumer Abdella.

**Resources:** Oumer Abdella.

**Supervision:** Francis Abuye.

**Visualization:** Francis Abuye.

**Writing – original draft:** Daniel Manore.

**Writing – review & editing:** Daniel Manore.

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
